# Comprehensive Analysis of the Chitinase Gene Family in Cucumber (*Cucumis sativus* L.): From Gene Identification and Evolution to Expression in Response to *Fusarium oxysporum*

**DOI:** 10.3390/ijms20215309

**Published:** 2019-10-25

**Authors:** Ezra S. Bartholomew, Kezia Black, Zhongxuan Feng, Wan Liu, Nan Shan, Xiao Zhang, Licai Wu, Latoya Bailey, Ning Zhu, Changhong Qi, Huazhong Ren, Xingwang Liu

**Affiliations:** 1Beijing Key Laboratory of Growth and Developmental Regulation for Protected Vegetable Crops, College of Horticulture, China Agricultural University, Beijing 100193, China; ezrabartholomew@hotmail.com (E.S.B.); kizzyb@hotmail.com (K.B.); fengzhongxuan@cau.edu.cn (Z.F.); liuwan@cau.edu.cn (W.L.); S20183172258@cau.edu.cn (N.S.); ZhangxiaoSY@cau.edu.cn (X.Z.); wulicai@cau.edu.cn (L.W.); Latoya.a.w.bailey@gmail.com (L.B.); renhuazhong@cau.edu.cn (H.R.); 2Changping Agricultural Technology Service Center, Beijing 102200, China; LZM629OK@163.com (N.Z.); cm89712413@163.com (C.Q.)

**Keywords:** *Cucumis sativus*, chitinase, *Fusarium oxysporum*, gene family, gene expression

## Abstract

Chitinases, a subgroup of pathogenesis-related proteins, are responsible for catalyzing the hydrolysis of chitin. Accumulating reports indicate that chitinases play a key role in plant defense against chitin-containing pathogens and are therefore good targets for defense response studies. Here, we undertook an integrated bioinformatic and expression analysis of the cucumber chitinases gene family to identify its role in defense against *Fusarium oxysporum* f. sp. *cucumerinum*. A total of 28 putative chitinase genes were identified in the cucumber genome and classified into five classes based on their conserved catalytic and binding domains. The expansion of the chitinase gene family was due mainly to tandem duplication events. The expression pattern of chitinase genes was organ-specific and 14 genes were differentially expressed in response to *F. oxysporum* challenge of fusarium wilt-susceptible and resistant lines. Furthermore, a class I chitinase, *CsChi23*, was constitutively expressed at high levels in the resistant line and may play a crucial role in building a basal defense and activating a rapid immune response against *F. oxysporum*. Whole-genome re-sequencing of both lines provided clues for the diverse expression patterns observed. Collectively, these results provide useful genetic resource and offer insights into the role of chitinases in cucumber-*F. oxysporum* interaction.

## 1. Introduction

Chitinases (EC 3.2.1.14) are glycosyl hydrolases (GH) that catalytic mechanism involves the hydrolysis of the β-1-4-linkage in the N-acetyl-D-glucosamine polymer of chitin, a major structural component of chitin-containing pathogens [1,2,3]. Chitinases are divided into two groups: endochitinase and exo-chitinase based on their cleavage and hydrolysis mechanisms [4,5]. Plants are devoid of chitin and it has been proposed that plant chitinases may play a role in defense response by degrading the chitin in the cell walls of invading pathogens [4,5,6]. Chitinases isolated from plants were reported to inhibit the growth of chitin-containing fungi, both in vitro [6,7] and in vivo [8,9], and over-expressed chitinases in plants confer resistance against several fungal pathogens [10,11,12,13,14]. Chitinases contains catalytic domains defining the two major GH families (GH18 and GH19) and are classified into seven classes (Class I–VII) [15,16]. GH18 chitinases (class III and V) are widely distributed in a variety of organisms, while GH19 chitinases (class I, II, IV, VI, VII) are found mainly in higher plants [17], and are responsible for the majority of chitinolytic activity within plants [4]. Despite shared chitinolytic activity, the GH18 and GH19 families do not share sequence similarity. The two families are clearly distinguished by their sequences and three-dimensional structures, indicating they are derived from different ancestral genes [4,18].

Cucumber is an economically important vegetable crop cultivated globally [19,20], which production is constrained by several biotic and abiotic stresses. Among biotic stresses, fusarium wilt caused by the soil-borne fungus, *Fusarium oxysporum* f. sp. *cucumerinum* J.H Owen, is a major threat to cucumber production worldwide [20,21,22,23]. Plants are equipped with a variety of defense mechanisms to protect themselves against pathogens, including the synthesis of pathogenesis-related (PR) proteins [24,25,26]. Chitinases, which comprise four of the PR protein families (PR-3/4/8/11), are either induced in direct response to pathogen elicitors or are constitutively expressed in tissues vulnerable to attack [4,27,28]. Davis et al. [29] reported that class I and class IV chitinases were induced after fusarium inoculation in *Ananas comosus*. Similarly, class IV and VII chitinase genes were up-regulated after *F. graminearum* infection in wheat [30]. Therefore, chitinases are good targets for defense response studies against fusarium species [18,31]. However, to date, the cucumber chitinase gene family has not been identified and systematically analyzed.

As a large gene family, it is important to elucidate the multiple functions of chitinase genes and the mechanisms responsible for the expansion of the gene family [15,32,33]. The availability of the draft cucumber genome [19] and transcriptome data [34], has promoted genome-wide identification of gene families and facilitated functional analysis studies. In this study, we reported a comprehensive genome-wide identification and analysis of the cucumber chitinase gene family. The response of chitinase genes to *F. oxysporum* infection was investigated in resistant and susceptible lines. We identified genome-wide nucleotide polymorphisms between resistant and susceptible lines, and these variations were functionally annotated in regions of putative chitinase genes. Our findings might provide effective gene resources for use in genetic analyses and improving *F. oxysporum* resistance in future cucumber-breeding programs.

## 2. Results

### 2.1. Genome-Wide Identification and Phylogenetic Analysis of Cucumber Chitinase Genes

In this study, the draft genome of *C. sativus* L. var. *sativus* cv. 9930 (ver. 2.0) [19] was used for genome-wide exploration of chitinase genes. A total of 28 putative chitinase genes were identified; each gene was annotated as *CsChiXX*, where *Cs* is the genus and species initials (*Cucumis sativus*) and *XX* is the sequential number of a gene in the genome. Biological characteristics of *CsChi* genes are listed in Table 1. Sequence analysis revealed that the lengths of the deduced chitinases vary from 132 (*CsChi8*) to 429 amino acids (*CsChi11*), with an average of 307 amino acids. The predicted molecular weights (MW) and the theoretical isoelectric points (pI) range from 14.80 kDa (*CsChi8*) to 48.53 kDa (*CsChi11*) and from 4.6 (*CsChi9*) to 9.53 (*CsChi18*), respectively. Of the 28 chitinases, 21 were predicted to localized in the secretory pathway, with 4 localizing in the chloroplast and 3 in the nucleus.

A total of 70 full-length chitinases sequences from *C. sativus*, *C. melo* and *Arabidopsis thaliana* (http://cucurbitgenomics.org;
https://www.arabidopsis.org) were used to construct a neighbor-joining phylogenetic tree (Figure 1A). On the basic of phylogeny and sequence homology, all 70 putative chitinase genes were grouped into two families, one belonging to GH19: class I, class II, class IV, and the other belonging to GH18: Class III and class V. Chitinases within the GH19 family were grouped into three distinct clades representing class I, class II and class IV. Whereas, chitinase within the GH18 family resolved into four distinct, well-supported clades. Class V chitinases were grouped into a single clade that shares little sequence similarity with the class III proteins. Class III chitinases were divided into three clades with distinct sequences. Class IIIa clade contained homologs of the human stabilin-1 interacting chitinase-like proteins (SI-CLPs). Class IIIb clade contained narbonin and nodulin-like proteins, which are globulin protein that lacks conserved chitinase catalytic residues and enzymatic activity [4], while class IIIc clade contained acidic endochitinase-like (AMCase-like) proteins.

### 2.2. Gene Structure and Conserved Motifs Analyses of Chitinase Genes

To study the structural conservation and diversity of chitinase genes, the exon-intron architecture and conserved motifs distribution was investigated. Most GH18 chitinase genes contained 1 or 2 exons, except for chitinases in the Class IIIa clade, which are SI-CLPs known for their large size and many exons (Figure 1B) [35]. Whereas, most GH19 chitinase genes had 3 exons. Closely related genes share high similarity in terms of the length, number of exons/introns and intron phases, and these features are highly conserved. Only 8 of the 17 *CsChi* genes belonging to the GH18 family contained introns, compared to 10 of 11 *CsChi genes* from the GH19 family. The absence of an intron region was most prevalent for chitinases in the Class IIIc clade. Most class I and II *CsChi* genes had 2 introns and uniform intron phases, while members of class IV had 1 intron. In addition, some intron loss or gain occurred within several evolutionary branches.

A total of 30 distinct motifs were identified and their composition were consistent with the results from the phylogenetic tree (Figure 1C). As expected, closely related members shared a common motif composition. Motifs 3–10 were displayed in the same order in most class I and II chitinases, with variations only at the N-terminal chitin-binding domain, which is lacking in class II chitinases. Members of class IV had motifs 1, 3, 4, 6, 7 and 14, except for AT3G04720 which contained motif 1 only. Motifs 15–22 were present in class V chitinase, however *CsChi6* lacked motifs 15, 18 and 22. The motifs composition of class III chitinases were divided into distinct three clades, which supports the results of the phylogenetic analyses, but also implied functional relevance. Furthermore, several distinct motifs were found in specific GH family, which supports the different origin of the GH18 and GH19 families.

### 2.3. Conserved Domains and Active Site Analysis of CsChi Genes

Cucumber chitinase sequences contained conserved domain structures in accordance with previously described plant chitinases [17,36]. Analysis of the amino acid sequences reveal the presence of the GH18 catalytic domain (PF00704) with the CHITINASE_18 (PS01095) active site signature in classes III and V chitinases (Figure 2A,B). Class V chitinases also possess additional signatures and were on average longer than the members of the other classes (Figure 2B). Analysis of the amino acid sequences of classes I, II and IV reveal the presence of the GH19 catalytic domain (PF00182) with one or two active sites signatures: CHITINASE_19_1 (PS00773) and/or CHITINASE_19_2 (PS00774). Additional, class I and class IV chitinases contained a chitin-binding domain with the CHIT_BIND_I_1 (PS00026) signature at the N- terminal (Figure 2C). Interestingly, we identified a class I-like gene, *CsChi24*, with an unusual structure containing two chitin-binding domains. Functional site analysis using ExPASy prosite [37], predicted that only 18 of the 28 cucumber chitinase proteins contained the active sites in their catalytic domain required for chitinolytic activity. The remaining 10 genes had mismatched amino acids in their active sites and were classified as putative chitinase-like proteins (CLP).

### 2.4. Chromosomal Distribution and Gene Duplication Analysis of CsChi Genes

To investigate the relationship between the genetic divergence and gene duplication of the cucumber chitinase gene family, their chromosomal locations were determined. The 28 *CsChi* genes were matched to chromosomes 1–6, with none identified on chromosome 7; the distribution of chitinase genes on each chromosome was uneven (Figure 3). Genes belonging to the same class were generally found in clusters. The *C. sativus* genome contained 25 genes that were orthologous to 7 *Arabidopsis* chitinases, indicating that most cucumber chitinases were duplicates (Appendix A). These duplicated genes were mainly found within tandemly arrayed gene clusters. Specifically, 5 clusters containing 20 cucumber chitinase genes belonging to classes I, II, III, and V were orthologous to just 3 *Arabidopsis* genes: *AtChiA* (AT5G24090), *AtChiB* (AT3G12500), and *AtChiC* (AT4G19800) (Appendix A). Furthermore, 4 pairs of paralogous chitinase genes (*CsChi1/CsChi6*, *CsChi12*/*CsChi18*, *CsChi13*/*CsChi19* and *CsChi10*/*CsChi23*) were located in region of segmental duplication (Figure 3).

### 2.5. Cis-Regulatory Elements in the Promoter of CsChi Genes

Chitinase genes have been reported to be transcriptionally regulated by several phytohormones and pathogen attack [18,38]. To elucidate the possible regulation mechanisms of *CsChi* genes, the *cis*-regulatory elements in the promoter sequence were predicted using PlantCARE [39] and PlantPAN 2.0 [40]. Twenty types of hormone and stress related *cis*-regulatory elements were detected (Appendix A): 10 hormone- responsive elements (ABRE, ABRE3a, ABRE4, AuxRR-core, ERE, GARE-motif, P-box, TCA-element, TGACG-motif/AS-1, and TGA-element) and 10 stress- responsive elements (DRE1, HD-ZIP I, HD-ZIP III, LTR, MBS, STRE, TC-rich repeats, VOZ1, W box, and WUN-motif). Twenty-two cucumber chitinase genes contained the hormone responsive ERE element (ethylene responsive), the ABRE element (abscisic acid responsive) was detected in 20, and the TCA-element (salicylic acid responsive) in 15. In addition, the stress responsive elements W-box (wounding and pathogen responsive) and STRE (stress responsive) was predicted in 17 chitinases each and the WUN-motif (wound responsive) in 16.

### 2.6. Spatial Expression Profiles of CsChi Genes

To gain insight into the functions of *CsChi* genes, RNA-seq data (NCBI accession number: PRJNA80169) from *C. sativus* L. var. *sativus* cv. 9930 were used to quantify the expression levels of chitinase genes in various tissues (leaf, stem, female flower, male flower, ovary, root, and tendril) [34]. The expression profiles of all 28 putative *CsChi* genes were visualized using heat maps (Figure 4). Four of the 28 genes were highly expressed in all tissues (*CsChi10*/*11*/*15*/*16*). Most chitinase genes exhibited consistent expression levels across all tissues with some genes not expressed or only weakly expressed in female flowers, ovaries, and stems. In contrast, several chitinase genes exhibited higher expression levels in their roots compared with the other tissues (*CsChi2*/*15*/*10*/*20*/*23*/*27*/*28*), suggesting that they may play a role against soil-borne pathogens.

### 2.7. Expression Analysis of CsChi Genes in Response to F. oxysporum

The expression patterns of chitinase genes have been shown to vary between resistant and susceptible plants in response to fungal infection [38,41]. To test whether the expression of *CsChi* genes respond to *F. oxysporum* infection and to gain further insight into the differential response of resistant and susceptible lines, we selected 18 putative enzymatically active *CsChi* genes for expression analysis by qRT-PCR. Two cucumber inbred lines (3461 = resistant; 3229 = susceptible), with different level of resistance to *F. oxysporum* were used (Figure 5A). Two weeks after inoculation the fusarium wilt disease symptoms was assessed (Figure 5B). No major wilt symptoms were observed in the resistant line 3461, but severe wilt symptoms were detected in the stems and leaves of susceptible line 3229.

The expression pattern of each *CsChi* gene was checked 3dpi, as it has been reported that genes encoding several PR proteins showed a peak in expression 1-3dpi when distressed with hemibiotrophic pathogens [42,43,44,45]. In line 3229, expression analysis revealed that most chitinase genes responded significantly to fungal infection (Figure 6). Specifically, 14 genes were significantly up-regulated after infection, while 2 genes (*CsChi26* and *CsChi28*) were down-regulated. In line 3461, the expression levels varied and 7 genes were up-regulated following fungal infection, 2 genes were down-regulated and no significant differences were observed in 9 genes. The expression of class I chitinase genes (*CsChi23* and *CsChi28*) were generally higher in resistant lines. Interestingly, *CsChi23* was constitutively expressed about 200-folds higher in line 3461 compared to line 3229 in mock infected plants.

### 2.8. Whole-Genome Re-Sequencing and Variation Analysis between Lines 3229 and 3461

Whole-genome re-sequencing of cucumber inbred lines 3229 and 3461 generated an average of 99,446,500 and 80,794,000 raw sequence reads, respectively. About 93% of these were clean reads, with about 91% of the bases being high-quality bases (quality score ≧ Q30). Overall, an average of 72,481,500 (78.95%) reads from line 3229 and 64,168,00 (84.94%) from line 3461 were uniquely mapped to the *C. sativus* L. (9930) reference genome at an average read depth of 31.7X and 31.35X, respectively (Appendix A). After single nucleotide polymorphism (SNP) and (insertion/deletion) InDel calling, 410,779 SNPs and 120,988 InDels were detected directly between both lines (Appendix A). Overall, 81% SNPs and 83% InDels were distributed in the intergenic regions (intergenic, upstream, and downstream), while about 19% SNPs and 17% InDels were observed in the genic regions (exonic, intronic and UTRs (Figure 7A,B). Furthermore, several large- effect variations that may have severe impacts on the integrity of encoded products and gene functions, such as, SNPs and InDels distributed in the initiation/termination codon and the InDels which caused frameshift were detected (Figure 7C).

### 2.9. DNA Polymorphisms in Region of Chitinase Genes

To understand the diverse expression patterns of chitinase genes observed between lines 3229 and 3461, we annotated all genetic variations found in the intergenic and genic regions of putative chitinase genes (Appendix A). A total of 690 genetic variations were identified between both lines, with about 80% (549) being SNPs and 20% (141) were InDels (Appendix A). In general, SNPs and InDels occurred more frequently in noncoding regions than in the coding regions. Of the total SNPs discovered, ∼35% were in genic regions with ∼20% being in exonic regions, while ∼11% of InDels were found in the genic regions with >1% being in exonic (Figure 7D,E). Interestingly, about 56% of all the genetic variations in chitinase genes was found in a gene cluster on chromosome 6 (*CsChi23*/*24*/*25*/*26*/*27*/*28*) which contains classes I and II chitinases (Appendix A). Furthermore, several large-effect variants in line 3229, including a splicing mutation and a frameshift-insertion was detected in *CsChi25* (Chr6: 26090784/ 26090779), while a stop-gain mutation was detected in *CsChi19* (Chr5: 3712267; Appendix A).

## 3. Discussion

Due to the constant threat of pathogens, plants have evolved sophisticated detection and response systems that decipher pathogen signals and induce appropriate defenses [24]. Inducible defense responses involve a variety of defense mechanisms including the production and secretion of antimicrobial compounds such as PR proteins and phytoalexins, cell wall modification, generation of reactive oxygen species, and programmed cell death or hypersensitive response at the site of infection to limit pathogen progression [24,25,26]. Chitinases, which comprise four of the PR protein families (PR-3/4/8/11), can inhibit fungal growth by catalyzing the hydrolysis of chitin, the main component of the pathogen cell wall, and are therefore good targets for defense response studies [18,31]. Most research on the identification and analysis of chitinase gene family have focused on *A. thaliana, Nicotiana benthamiana, Oryza sativa* and *Zea mays*, whereas studies are limited on non-model plants like *C. sativus* [46]. A total of 28 putative chitinase genes were identified in the cucumber genome. Based on their protein sequences, all cucumber chitinases are endochitinases (EC 3.2.1.14) and are classified into 5 different classes (I–V). Here, we undertook detailed bioinformatic and expression analysis of the cucumber chitinases genes to understand their roles in cucumber- *F. oxysporum* interaction

Due to the great diversity of enzymes that hydrolyze or bind chitin, chitinases have been proposed as good examples of gene evolution related to gene duplication [47]. Chromosomal localization and gene duplication studies indicated that the cucumber genome contains 5 tandemly arrayed gene clusters with 20 chitinase genes orthologous to 3 *Arabidopsis* genes (Appendix A). Therefore, >71% of cucumber chitinase genes originated from tandem duplication (Appendix A). Over the process of biological evolution, duplicated genes may acquire new structures and functions, resulting in chitinase with diverse binding, substrate, and pathogen specificity that are uniquely expressed in different tissues or under different stresses [48,49]. Interestingly, we identified a novel class I-like chitinase gene (*CsChi24*) that had an unusual structure containing two chitin-binding domains (Figure 2C). *CsChi24* shared a close relationship with *CsChi23* (class I) and *CsChi25* (class II), which contained one and zero chitin-binding domain, respectively. Hardt and Laine [50], indicated that mutations of active site residues in the chitin-binding domain, caused by unequal crossing-over, can have profound effects on the expression and substrate specificity of chitinase genes. We hypothesize that structural differences in the chitin-binding region of *CsChi24* may affect its binding specificity leading to functional diversification. Therefore, functional differentiation may be the mechanism driving the evolution and expansion of the cucumber chitinase gene families.

Gene structure and conserved motif distribution can provide further evidences for the evolutionary relationships of gene families. Intron positions are highly conserved in orthologous genes across species, and intron gain and loss can increase the complexity of gene organization [51]. Most cucumber chitinase genes within the same class had uniform exon-intron organization and intron phases. However, some intron loss or gain occurred within several evolutionary branches (Figure 1B). The exon-intron architecture of chitinase genes is also known to mediate the stress responses. Generally, the structures of stress-related genes contain few introns [51]. In this study, 25 of the identified 28 *CsChi* genes had two or fewer introns (Figure 1B). This adds to the conception that longer and several introns can prolong transcription, whereas fewer introns are present in genes that are rapidly regulated during stress [51]. The absence of an intron region was most prevalent in class IIIb and class IIIc chitinases gene. Similar chitinase genes structures were observed in *B. rapa* and found in other stress-related gene families [52,53]. Moreover, a similar motif composition was found in each group, suggesting functional relevance, while diverse motif composition implies functional diversification among different groups (Figure 1C).

Although plants have no chitin in their cell walls, they produce many forms of chitinases, including cellular and secretory chitinases, that play key roles in multiple physiological processes. Most cucumber chitinases were predicted to localize in the apoplast and involved in the secretory pathway (Table 1). The plant apoplast is integral for intercellular signaling, transport and plant-pathogen interactions [54]. Apoplastic chitinases are usually induced immediately after pathogen infection and can directly block the growth of the hyphae at the intercellular space [55,56]. Also, apoplastic chitinases are known to act directly or indirectly with pathogen elicitors, which triggers downstream defense pathways [57]. These results indicate that apoplastic *CsChi* genes may participate in chitinase-pathogen interaction.

Chitinases are transcriptionally regulated by several *cis*-regulatory elements, including hormone and stress *cis*-regulatory elements [18,39,58]. The hormone-responsive element ERE (ethylene-responsive) was found in the promoter region of 22 *CsChi* genes, indicating that the expression of most chitinase genes might be induced by ethylene (Appendix A). Several studies also identified ethylene as a key hormone involved in chitinase gene induction [59,60]. Other hormone responsive elements (ABRE, TCA, and TGACG-motif) predicted to regulate transcription of chitinase genes [61,62,63] were also identified in the promoter of most *CsChi* genes (Appendix A). Likewise, several known stress responsive elements (W-box, STRE, and WUN-motif) predicted to mediate pathogen- and/or elicitor-inducible transcription of chitinases [39,58,64,65,66] were also identified. These results indicate that *CsChi* genes may be transcriptionally regulated by hormone and stress *cis*-regulatory elements and may be induced by pathogen elicitors.

Cucumber chitinase gene showed differential expression patterns in the seven different cucumber tissues. Several genes were highly expressed in all tissues, these include developmentally-regulated CLPs: *CsChi16* (CLP-1), *CsChi15* (CLP-2) and *CsChi11* (SI-CLP), and *CsChi10*, an ortholog of AT4G01700 known to be constitutively expressed in all tissues in *Arabidopsis* [18,55]. Chitinase genes were reported to be highly expressed in roots and flowers in several plant species [18,67]. This study showed similar expression patterns, with several chitinase genes being highly expressed in the roots and male flowers (Figure 5). In the roots, GH19 *CsChi* genes exhibited higher expression levels indicating that they may be involved in plant defense against soil-borne pathogens such as *F. oxysporum*. Samac et al. [55] also reported that the GH19 family of chitinases in *Arabidopsis* were predominantly expressed in roots and may play a role in stress response.

PR proteins, including chitinases are either silent or constitutively expressed at low levels in plants during the absence of pathogens, but are significantly induced in response to pathogens infections [68,69,70,71,72]. In agreement with previous reports, the expressions of *CsChi* genes showed both constitutive and induced expression patterns. Here, at least 14 *CsChi* genes were induced after *F. oxysporum* inoculation (Figure 6). Class I chitinase genes were highly expressed in resistant line 3461, whereas class III, IV and V chitinases were upregulated in line 3229. Similar gene expressions were also reported by Adhikari et al. [43] and Xin et al. [73] in wheat host following *Mycosphaerella graminicola* and powdery mildew infection. The chitinase gene family has many diverse functions that are regulated by numerous stimuli such as abiotic and biotic stresses as well as various phytohormones. Polymorphisms in chitinase genes and regulatory regions are also known to affect chitinase activity and expression [55,64,65]. We suspect that the varying expression patterns observed between lines 3229 and 3461 may be related to the functional diversity of cucumber chitinase genes and evolutionary differences of both lines. Several class I chitinases (*CsChi23* and *CsChi28*) were induced in resistant line 3461. These are PR-3 proteins that are known to deter fungal pathogens by acting on chitin in their cell walls [55]. Interestingly, many chitinase genes were not induced upon *F. oxysporum* inoculation in line 3461. These include chitinase isoforms: *CsChi5* (class IV, a homolog of carrot EP3), and *CsChi4*, *CsChi9* and *CsChi22* (class III, 2S globulin/ Narbonin homologs) that are known for their roles in plant embryogenesis and as storage protein found in seeds [4,18]. These findings further confirm that cucumber chitinase genes have diverse functions.

Xin et al. [73] suggests that some genes with pathogen-defense functions are constitutively expressed at high levels in resistant hosts and may act as a constant barrier to pathogen attacks. In this study, the expression patterns of *CsChi* genes in both resistant and susceptible lines were similar under control conditions, except for *CsChi23*, which was constitutively expressed about 200-folds higher in line 3461 (Figure 6). Takenaka et al. [74] reported that *Arabidopsis* class I chitinase gene *AtchiB*, a homolog of *CsChi23*, was also constitutively expressed at a high level. Class I chitinases have been extensively studied for its defensive roles and are known to inhibit hyphal growth of several fungal species and degrades chitin more rapidly than other classes of chitinase [75,76,77,78]. The constitutive expression of rice class I chitinase, *Os*Chia1b, in several commercial crops also resulted in enhanced resistance to fungal diseases [79,80]. After *F. oxysporum* infection, *CsChi23* was highly upregulated in both lines. This indicates that *CsChi23* expression is induced by *F. oxysporum*. We speculate that the high constitutive expression of *CsChi23* in resistant cucumber lines may play a critical role in building a basal defense and a rapid immune response against fungal attacks. However, further research is needed to validate the antifungal role of *CsChi23* in *C. sativus*-*F. oxysporum* interaction.

The location and type of DNA polymorphism have a major influence on gene expression [81]. A large proportion of DNA polymorphisms are in the intronic and intergenic regions, which hinders their functional evaluation. The promoter region located upstream of the initiation site encompasses *cis*-regulatory sites that play an important role in regulating gene expression. Polymorphisms in the promoter region may alter the transcription factor binding sites, thus affecting gene expression [82]. In this study, about 84% of the SNPs and InDels observed in putative *CsChi* genes between lines 3229 and 3461 occurred in the non-coding regions, with about 21% occurring in the upstream (2 kb) promoter region (Figure 7D,E). Chorley et al. [83] suggests that in some cases, a natural binding site created or abolished by a regulatory polymorphism can account for observed differences in gene expression. We speculate that the diverse expression of *CsChi* genes between lines 3229 and 3461 may be caused by such polymorphisms in the promoter region. Interestingly, about 56% of all genetic variations in chitinase genes were found in a gene cluster on chromosome 6 (CsChi23/24/25/26/27/28) which contains classes I and II chitinases (Appendix A). These polymorphisms may have a major effect on gene structures, functions, and expression. About 20% of the SNPs occurred in the coding region and several large-effect variants were detected which may affect chitinase activity. Furthermore, a large proportion of polymorphisms was found in the promoter region of *CsChi23*. To validate these polymorphisms, the putative *CsChi23* promoter fragment (from 2 kb upstream of the ATG initiation codon) of line 3461 and 3229 was cloned and sequenced (Appendix A). Sequence analysis revealed multiple sequence differences, including 21 SNPs and 3 InDels, that abolish the HD-Zip III transcription factor motif, the P-box motif (gibberellin-responsive element) and the TGA motif (early auxin-responsive element) in line 3229. These polymorphisms might explain the constitutive expression of *CsChi23* in line 3461. However, further promoter analysis of *CsChi23* between lines 3229 and 3461 is needed to determine the effect of DNA polymorphisms.

## 4. Materials and Methods

### 4.1. Sequence Acquisition and Identification of Cucumber Chitinase Genes

To identify chitinase genes, amino acid sequences from *Arabidopsis* and melon chitinases were BLASTP (https://blast.ncbi.nlm.nih.gov/Blast.cgi) queried against cucumber. NCBI-CDD [84] and HMMER web server: 2018 update [85] were used to analyze the conserved domains of all non-redundant sequences. Hidden Markov models (HMMs) profiles of the chitinase catalytic domains from GH18 (PF00704) and GH19 (PF00182) were obtained from the Pfam database (Pfam 32.0) [86]. EMBL-EBI InterPro [87] was used to validate the identified genes. ExPASy prosite [37] was used to analyze all putative chitinase genes for the presence of the catalytic residues required for chitinolytic activity. The gene and coding sequence (CDS) (bp), number of amino acids and chromosome location of cucumber chitinase genes were obtained from Cucurbit Genomics Database (http://cucurbitgenomics.org). The molecular weight (kDa) and isoelectric point (pI) of each chitinase protein were analyzed using ExPASy’s Compute pI/Mw tool [88]. Subcellular localization was predicted using BaCello [89] and signal peptide cleavage sites were predicted using SignalP 4.1 [90].

### 4.2. Phylogenetic Tree, Gene Structures and Conserved Motifs Analyses of Chitinase Genes

To study evolutionary relationships, full-length amino acid sequences of 70 chitinase proteins, from *C. sativus*, *C. melo* and *A. thaliana* were first aligned using ClustalW [91] with default settings to construct an unrooted neighbor-joining (NJ) phylogenetic tree with Poisson correction using MEGA X [92]. Bootstrap analysis was employed using 1000 replicates. The pair-wise gap deletion mode was used to ensure that the more divergent C-terminal domains could contribute to the topology of the NJ tree. The CDS and corresponding genomic sequence of each chitinase genes were used to predict the exon–intron structures using GSDS v2.0 [93]. The MEME software v. 5.0.5 [94] was used to identify conserved motifs in each chitinase protein sequences according to the following parameters command: -protein -oc -nostatus -mod zoops -nmotifs 30 -minw 6 -maxw 50 -objfun classic. Multiple sequence alignment of cucumber chitinases amino acid sequences from was conducted using DNAMAN software v6.0 (Lynnon BioSoft, Quebec, Canada).

### 4.3. Chromosomal Location and Identification of Homologous A. thaliana Members

All identified cucumber chitinase genes were mapped to six chromosomes and visualized by Mapchart v. 2.32 [95] using base pair start position information from the cucurbit genomics database. Clusters of chitinase genes were determined based on previously defined criteria by Christie et al. [96]. Duplications between the *CsChi* genes were identified using the cucurbit genomics database. To identify orthologous cucumber chitinase genes in *Arabidopsis*, a phylogeny-based method was used and a BLAST-based reciprocal best hit method was used to verify the putative orthologous genes. The Circoletto tool [97] was used to plot sequence similarity using the pairwise alignment of cucumber and *Arabidopsis* chitinases.

### 4.4. Cis-Regulatory Elements Analysis

To predict the *cis*-regulatory elements of each cucumber chitinase genes, we obtained the nucleotide sequences of the promoter regions (1.5 kb upstream of the initiation codon) from the draft genome of *C. sativus* L. var. *sativus* cv. 9930 (ver. 2.0). PlantCARE [39] and PlantPAN 2.0 [40] were used to predict c*is*-regulatory elements.

### 4.5. Gene Expression Analysis

To determine the expression patterns of cucumber chitinase genes in various tissues (leaf, stem, female flower, male flower, ovary, root, and tendril) we obtained RNA-Seq data (NCBI accession number: PRJNA80169) [34]. Tissues were obtained from *C. sativus* var. *sativus* line 9930. The sequencing data is obtainable with the accession number SRA046916 in the Sequence Read Archive (SRA) at NCBI. The absolute transcript abundance values obtained for all chitinase genes were derived based on the alignments using Bowtie 2 [98] and HISAT2 [99] and then normalized using EXPANDER7 [100] to RPKM (reads per kilobase of exon per million mapped fragments).

### 4.6. Plant Material and Fusarium Infection Assay

Two cucumber inbred lines, 3461 and 3229, which exhibited resistance and susceptibility to *F. oxysporum,* respectively, were used in the present study. Seeds were surface sterilized and sown in autoclaved soil (50/50 nutritive soil/vermiculite). All plants were grown at 28 °C/22 °C under 16-h light/8-h dark conditions. The fungus, *F. oxysporum* f. sp. *cucumerinum* race 3, obtained from China Agricultural University, Beijing, P. R. China, was used as inoculating pathogen. For infection assays, the inoculum was prepared as described by Wang et al. [22], with some modifications. Cucumber seedlings at two-true leaves stage were root dip-inoculated with a conidial suspension (5 × 10^5^ spores/mL), or mock-inoculated with sterile water, after which seedlings were transplanted into 250mL plastic pots containing autoclaved soil. Development of disease symptoms was followed over time. Disease severity for each plant was scored on a scale of 0–4, only plants with an average score of 1 were considered resistant. The experiment included 6 plants per replicate with two replications. Samples were collected at 3 days post inoculation (dpi) for expression analysis and the severity of the disease was assessed 14 dpi [20]. All tissues were immediately frozen in liquid nitrogen and stored at −80 °C.

### 4.7. RNA Extraction and Quantitative Real-Time Reverse Transcription PCR (qRT-PCR) Analysis

Total RNA was extracted using the Quick RNA plant isolation kit (Beijing Yueyang Biotechnology Ltd., Beijing, China), and cDNA was synthesized using the OneScript^®^ cDNA Synthesis kit with AccuRT Genomic DNA Removal (Applied Biological Materials Inc, Vancouver, Canada). qRT-PCR was performed in 96-well plates with an Applied Biosystems 7500 real-time PCR system (Applied Biosystems, Foster city, CA, USA) using the One-Step BrightGreen qRT-PCR-Low ROX kit (Applied Biological Materials Inc, Vancouver, Canada). Primers were designed using Primer3Plus, and primer specificity was evaluated with NCBI primer-BLAST against the cucumber genome. The specificity of the amplified product was confirmed by a melting curve. Three biological replicates and 3 technical replicates were performed for each combination of cDNA samples and primer pairs. The cucumber β-actin served as the internal control gene [101]. The relative expression of target genes was analyzed by comparative CT method [102]. The cucumber chitinase qRT-PCR primers are listed in Appendix A.

### 4.8. DNA Extraction, Whole-Genome Resequencing and Variation Analysis

Genomic DNA from young leaves was extracted using the Plant Genomic DNA Rapid Extraction kit (Aidlab Biotechnologies, Beijing, China). The concentration and quality of DNA were determined using a NanoDrop2000 Spectrophotometer (Thermo Fisher Scientific, Waltham, Massachusetts, USA). DNA libraries for Illumina sequencing were constructed according to the manufacturer’s specifications (Illumina, San Diego, CA, USA). Paired-end sequencing libraries with an insert size of 300-400 bp were sequenced on an Illumina HiSeq 2500 sequencer by a commercial service (ORI-GENE, Beijing, China). Paired-end resequencing reads from lines 3229 and 3461 were aligned against the *C. sativus* L. (9930) reference genome using the MEM algorithm of Burrows–Wheeler Aligner [103] (BWA v0.7.5a-r405), after trimming of low-quality reads by using Trimmomatic [104]. Picard (release 2.0.1, https://broadinstitute.github.io/picard/) was used to eliminate PCR duplicates and to realign InDel regions. SNP and InDel calling for each sample were performed using Genome Analysis Toolkit (GATK) v4.1.2 (https://software.broadinstitute.org/gatk/). The variation detection followed the best practice workflow recommended by GATK [105]. Initially, variants from lines 3229 and 3461 relative to the reference genome were called separately, then merged and compared to each other. Variants located in the regions of chitinase genes were then identified and analyzed.

### 4.9. Validation of Variations

The sequences of selected variations located in the regions of chitinase genes were used as template for primer design (Appendix A). The variation regions were amplified by PCR using genomic DNA extracted from lines 3229 and 3461. The sequences were confirmed by repeated sequencing by Sanger sequencing method. The assembled sequences were aligned to the reference genome sequences to validate the variations in all the selected regions using ClustalW [91].

## 5. Conclusions

In summary, we performed a complete genome-wide identification and analysis of the chitinase gene family in cucumber and generated expression data of both resistant and susceptible lines after *F*. *oxysporum* inoculation. Twenty-eight chitinases were identified in the *C. sativus* genome and their key structural features were examined. This provides valuable information for further elucidation of the evolution and expansion of chitinase gene family. The high constitutive expression of a class I chitinase in resistant line 3461, *CsChi23*, may play a key role in enhancing fungal resistance. Moreover, whole-genome re-sequencing of resistant and susceptible lines provided clues for the diverse expression patterns observed. The information obtained from this study provides new insights into potentials roles of cucumber chitinases in plant defense and gives valuable gene resources for improving *F. oxysporum* resistance in future cucumber-breeding programs.

## Figures and Tables

**Figure 1 ijms-20-05309-f001:**
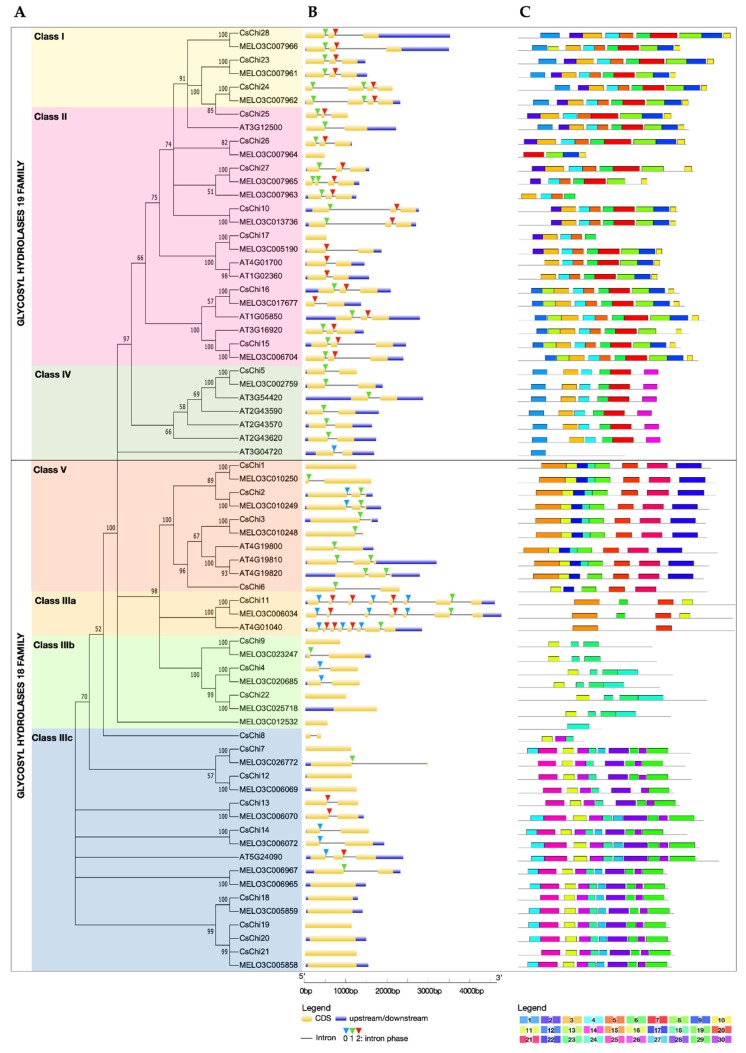
Phylogenetic and structure analysis of chitinase genes from *C. sativus*, *C. melo* and *A. thaliana*. (**A**) Phylogenetic tree built using the neighbor-joining (NJ) method in MEGA X. Boot strap values are from 1000 replications. The roman numerals (I–V) representing each chitinase gene class. The numbers at the nodes represent bootstrap percentage values. (**B**) Schematic representation of exon-intron structure of chitinase genes built using GSDS 2.0. Lengths of exons and introns of each gene are exhibited proportionally. For all genes, black lines represent introns, yellow boxes represent exons and purple boxes represent untranslated regions (UTRs). Intron phases 0, 1 and 2 are represented by blue, green and red triangles, respectively. (**C**) Schematic representation of conserved motifs of chitinase genes from *C. sativus*, *C. melo* and *A. thaliana*. The conserved motifs in chitinase proteins were identified by MEME software. Grey lines represent the non-conserved sequences, and each motif is indicated by a colored box numbered on the bottom of the figure. The length of motifs in each protein is presented proportionally.

**Figure 2 ijms-20-05309-f002:**
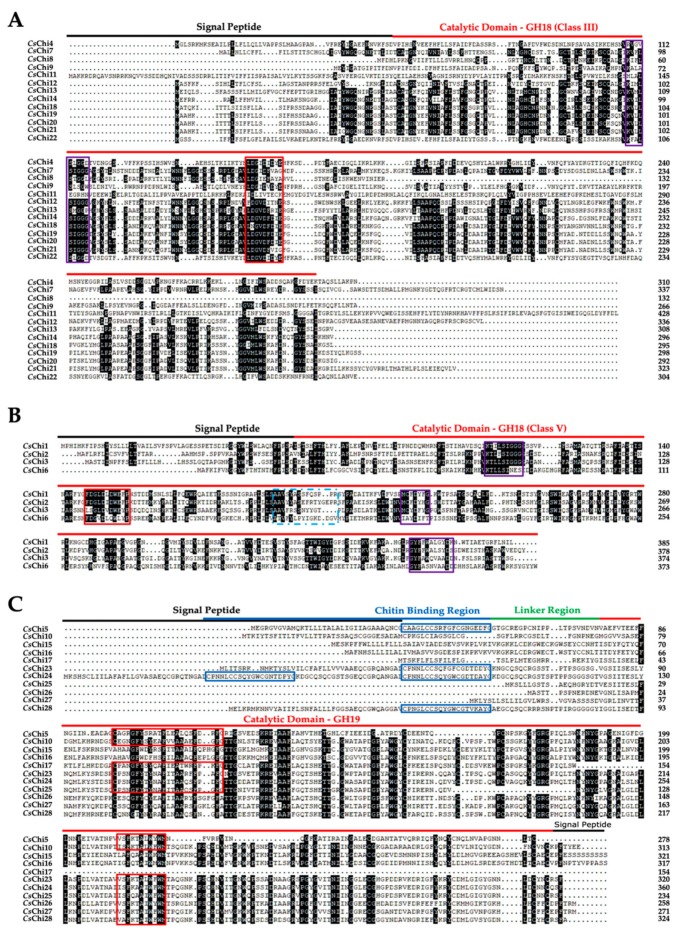
Multiple sequence alignment of the conserved domains of cucumber chitinase. (**A**) Sequence alignment of GH18 class III chitinases. Shaded amino acids are 50–100% homologous. (**B**) Sequence alignment of GH18 class V chitinases. Shaded amino acids are 75–100% homologous. (**C**) Sequence alignment of GH19 (class I, II and IV) chitinases. Shaded amino acids are 75–100% homologous. Amino acid sequences were aligned using DNAMAN software (ver. 6.0). Black and red lines over sequences indicate signal peptides and catalytic domain. While chitin-binding and linker regions are indicated by blue and green line over sequence. Purple boxed residues are conserved. Blue box (dash line) = PS00225, red box = catalytic domain and blue boxes = chitin-binding domain.

**Figure 3 ijms-20-05309-f003:**
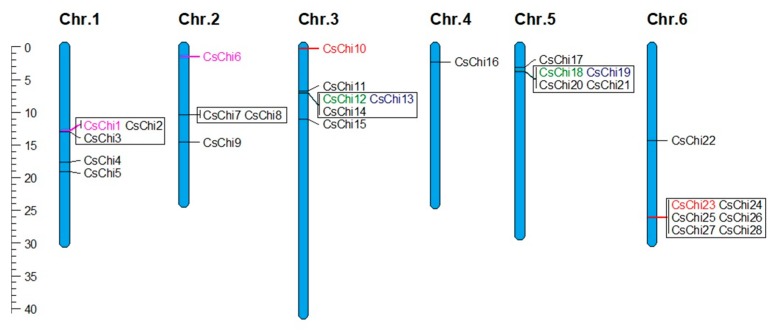
Chromosomal location and duplication of chitinase genes in cucumber. The chromosome size is indicated by its relative length using the information from the cucumber 9930 draft genome ver. 2.0 [19]. Black boxes indicated tandem duplication genes. The paralogous genes were written with same colors. Locations are mapped according to base pair start positions. The chromosome number is indicated above each bar and the scale bar on the left is in megabase (Mb).

**Figure 4 ijms-20-05309-f004:**
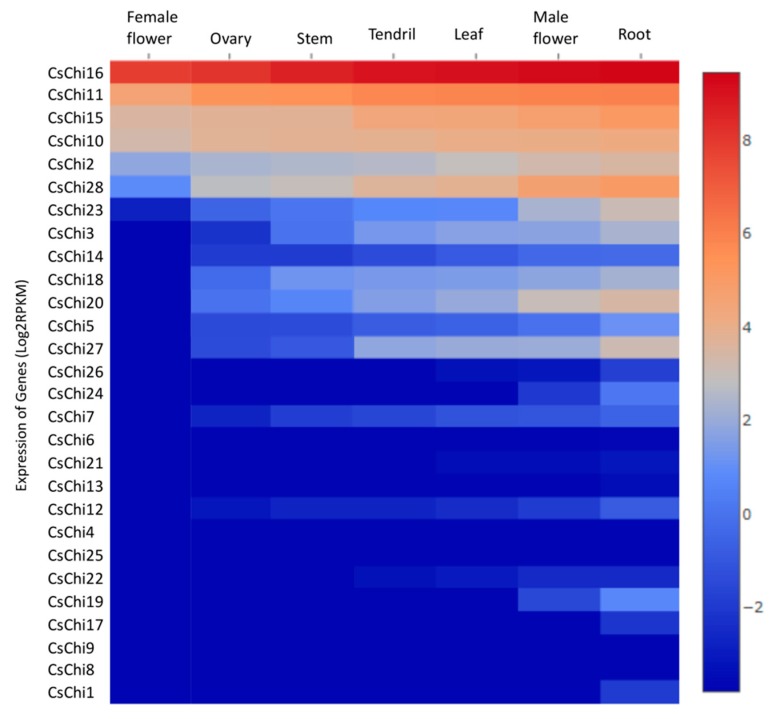
Heat map representation of chitinase genes expression in different cucumber tissues. The expression data was converted with Log2 (RPKM) to calculate gene expression levels. Gene expression differences are shown in the colors indicated in the scale. The RNA-Seq data used here could be downloaded from http://www.ncbi.nlm.nih.gov/bioproject/PRJNA80169/.

**Figure 5 ijms-20-05309-f005:**
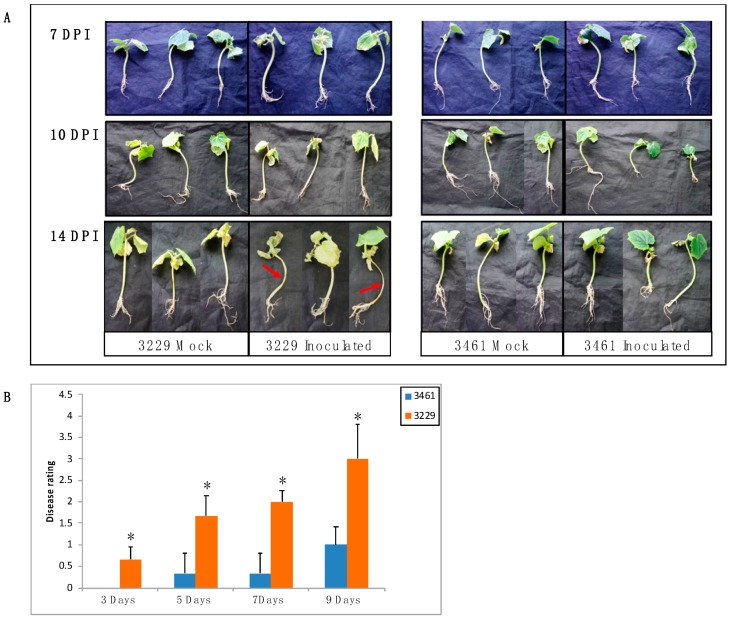
Fusarium wilt infection assay. (**A**) Disease symptoms of fusarium wilt in cucumber inbred lines 3229 and line 3461. The roots were inoculated with *F. oxysporum* f. sp. *Cucumerinum* Owen race 3. Mock plants were treated with distilled water. Red arrows show severe wilting in the stem. (**B**) Average disease symptom rating of both lines was scored using a disease rating scale (0–4). Values are shown as means ± SD (*n* = 6). (*, *p* < 0.05; Student’s *t*-test).

**Figure 6 ijms-20-05309-f006:**
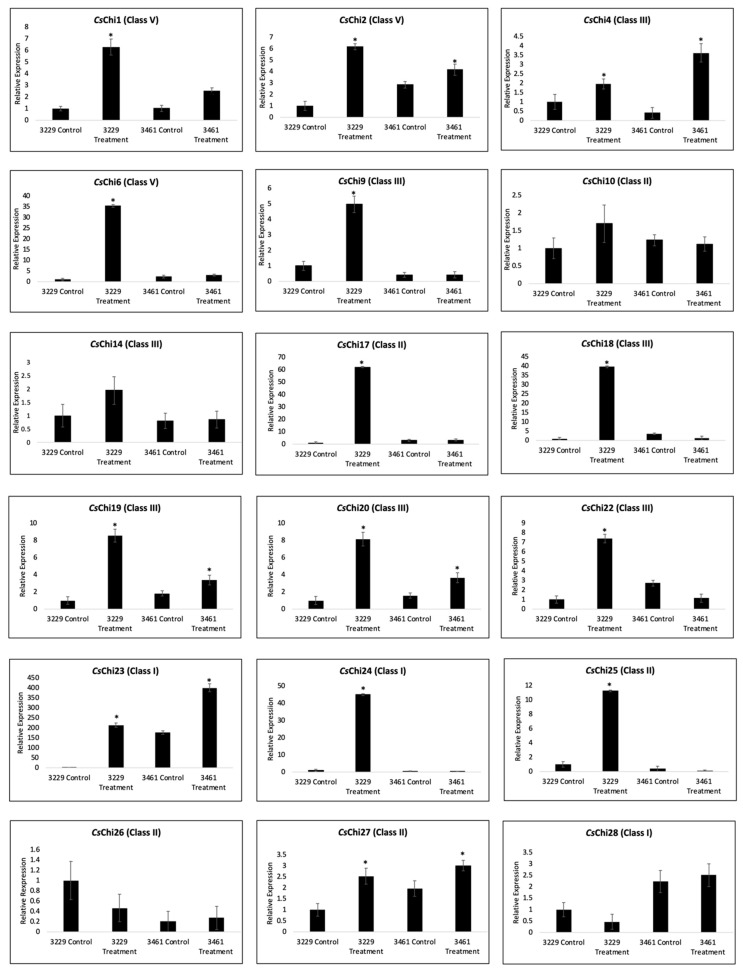
Expression patterns of 18 chitinase genes in *C. sativus* lines 3229 (susceptible) and 3461 (resistant) after infection with *F. oxysporum* f. sp. *cucumerinum* Owen race 3. Data were normalized to the expression level of β-actin. All data points are the means ± SE (*n* = 3). (*, *p* < 0.05; Student’s *t*-test).

**Figure 7 ijms-20-05309-f007:**
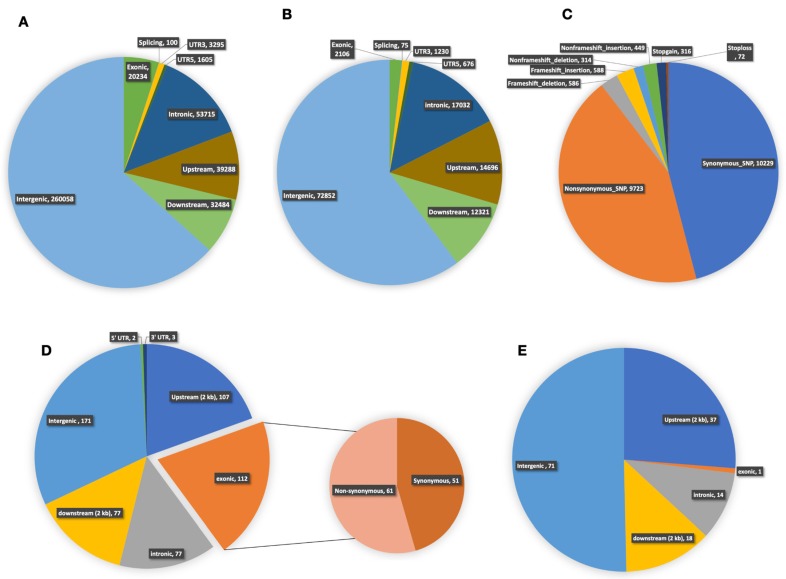
Annotation and distribution of single-nucleotide polymorphisms (SNPs) and insertions and deletions (InDels). Distribution of (**A**) SNPs; and (**B**) InDels in different genomic regions. (**C**) Annotation of total large-effect SNPs and InDels in different genic regions. Distribution of (**D**) SNPs; and (**E**) InDels in different intergenic and genic regions of chitinase genes from *C. sativus* lines 3229 vs 3461. The number of synonymous and non-synonymous SNPs detected within the CDS region has also been shown. Distribution of SNPs and InDels were 3229 vs 3461.

**Table 1 ijms-20-05309-t001:** Detailed information about 28 predicted cucumber chitinase genes.

Gene Name	Gene ID	Class	Gene Position		CDS (bp)	Size (aa)	MW (kDa)	pI	Localization	
			Start	End (+/−)					Signal P	BaCello
*CsChi1*	Csa1G267220	V	12946374	12947531 (+)	1158	385	42.11724	4.95	S.P	S.P
*CsChi2*	Csa1G267230	V	12949093	12950626 (−)	1137	378	42.499	7.73	S.P	S.P
*CsChi3*	Csa1G267240	V	12954742	12956249 (−)	1125	374	40.43333	8.86	S.P	S.P
*CsChi4*	Csa1G496300	III	17555204	17556387 (+)	933	310	34.62323	6.14		C
*CsChi5*	Csa1G534750	IV	19147451	19148640 (+)	837	278	30.03272	4.79	S.P	S.P
*CsChi6*	Csa2G008760	V	1514322	1516483 (−)	1125	374	42.66634	6.87		C
*CsChi7*	Csa2G193360	III	10356474	10357487 (−)	1014	337	36.81949	5.09	S.P	S.P
*CsChi8*	Csa2G193370	III	10361066	10361589 (−)	399	132	14.80392	8.84	S.P	S.P
*CsChi9*	Csa2G302200	III	14586790	14587590 (+)	801	266	29.79955	4.6		N
*CsChi10*	Csa3G002420	II	304482	307249 (+)	942	313	33.64859	5.14	S.P	S.P
*CsChi11*	Csa3G119680	III	6826381	6830851 (+)	1290	429	48.53536	6.91		C
*CsChi12*	Csa3G120470	III	6993545	6994578 (+)	1011	336	37.34421	5.85	S.P	S.P
*CsChi13*	Csa3G120480	III	6996557	6997717 (+)	927	308	34.25192	8.54	S.P	S.P
*CsChi14*	Csa3G120500	III	7006203	7007553 (+)	891	296	31.94549	8.69	S.P	S.P
*CsChi15*	Csa3G166220	II	10980690	10983104 (−)	969	322	35.63844	6.2	S.P	S.P
*CsChi16*	Csa4G017110	II	2283413	2285478 (−)	957	318	35.41338	6.23	S.P	S.P
*CsChi17*	Csa5G128240	II	3140161	3140625 (+)	405	154	17.33685	8.61	S.P	S.P
*CsChi18*	Csa5G139730	III	3709013	3710042 (+)	888	295	31.12939	9.53	S.P	C
*CsChi19*	Csa5G139740	III	3711563	3712459 (−)	897	298	31.92364	4.12	S.P	S.P
*CsChi20*	Csa5G139760	III	3715029	3716187 (−)	879	292	30.76542	4.38	S.P	S.P
*CsChi21*	Csa5G139770	III	3718639	3719610 (−)	972	323	34.47814	5.52	S.P	S.P
*CsChi22*	Csa6G301040	III	14333727	14334641 (−)	915	304	34.10921	5.61	S.P	S.P
*CsChi23*	Csa6G507520	I	26077903	26079303 (−)	963	320	34.829	8.4	S.P	S.P
*CsChi24*	Csa6G508020	I	26082160	26084273 (−)	1083	360	38.73501	5.71	S.P	S.P
*CsChi25*	Csa6G508520	II	26090192	26091124 (−)	705	234	25.79373	5.78		N
*CsChi26*	Csa6G509020	II	26095683	26096793 (+)	777	258	29.19341	8.78		N
*CsChi27*	Csa6G509030	II	26098005	26099490 (+)	818	271	29.96452	9.44	S.P	S.P
*CsChi28*	Csa6G509040	I	26100924	26104416 (+)	975	324	35.8648	6.31	S.P	S.P

S.P, Secretory pathway; C, Chloroplast; N, Nucleus; CDS, the coding sequence of a gene; aa, amino acid; +/−, the positive (sense) and negative (antisense) strand of DNA.

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
