# Peer review of "Comprehensive Analysis of the Chitinase Gene Family in Cucumber (Cucumis sativus L.): From Gene Identification and Evolution to Expression in Response to Fusarium oxysporum"

_ijms, 2019, doi:10.3390/ijms20215309_

Round 1
Reviewer 1 Report
The manuscript “Comprehensive Analysis of the Chitinase Gene Family in Cucumber (Cucumis sativus L.): From Gene Identification and Evolution to Expression in Response to Fusarium oxysporum” is well written, well organized and is very interesting to read.
The only shortcoming is the quality of the figures, particularly Figures 1 and 2. Even enlarging the image, the legibility is low, and it is impossible to read the legends.
Some minor comments:
Results
Please check values in line 72.:
‘Sequence analysis revealed that the lengths of the deduced chitinases vary from 132 (CsChi17)’ – is not CsChi8?
Also in Line 73: ‘the theoretical isoelectric points (pI) range from 17.33 kDa (CsChi17)’ – is not 14.80392 in CsChi8?
Line 149:
Fig. 2 - Why is the work of Passarinho and de Vries [17] cited in figure legend? If it refers to the adopted code of colours it should be stated clearly.
Arabidopsis – in italic font (lines 157, 160, 271, 321, 326, 345, 376, 405, 408)
Line 254
Figure 7 - To me is not clear how this figure illustrates the comparison between the two lines as referred in the legend (Annotation and distribution of SNPs and InDels between C. sativus lines 3229 and 3461.)
Discussion
Line 301 – I don’t understand why Table 1 is here referred.
To follow the discussion (e.g. ln 331 – 333), it would be helpful to have the information about the class of each chitinase gene in the Figures (Fig. 6 in this case).
Ln. 346 ‘Class I chitinases have been extensively studies for’ - check English
Author Response
Response to Reviewer 1 Comments
Point 1: The only shortcoming is the quality of the figures, particularly Figures 1 and 2. Even enlarging the image, the legibility is low, and it is impossible to read the legends.

Response 1: The image quality of all the figures was increased within the text and high-quality TIFF files (resolution 300 dpi) were uploaded along with the manuscript.
Point 2: Please check values in line 72.:
‘Sequence analysis revealed that the lengths of the deduced chitinases vary from 132 (CsChi17)’ – is not CsChi8?
Also in Line 73: ‘the theoretical isoelectric points (pI) range from 17.33 kDa (CsChi17)’ – is not 14.80392 in CsChi8?
Response 2: These values were corrected.
Point 3: Line 149:
Fig. 2 - Why is the work of Passarinho and de Vries [17] cited in figure legend? If it refers to the adopted code of colours it should be stated clearly.
Response 3: The work of Passarinho and de Vries [17] was cited for its contribution in identifying the conserved domains and signatures of chitinase genes in Arabidopsis. Therefore, the citation was removed from the figure caption but kept in-text.
Point 4: Arabidopsis – in italic font (lines 157, 160, 271, 321, 326, 345, 376, 405, 408)
Response 4: Adjusted to italic font.
Point 5: Line 254
Figure 7 - To me is not clear how this figure illustrates the comparison between the two lines as referred in the legend (Annotation and distribution of SNPs and InDels between C. sativus lines 3229 and 3461.)
Response 5: “Initially, variants from lines 3229 and 3461 relative to the reference genome were called separately, then merged and compared to each other.”
The figure caption was adjusted to show that the distribution of SNPs and InDels were 3229 vs 3461.
Point 6: Line 301 – I don’t understand why Table 1 is here referred.
Response 6: “Most cucumber chitinases were predicted to localize in the apoplast and involved in the secretory pathway”. Table 1 was referred to because it contained data about the predicted localization of cucumber chitinase proteins using SignalP and Bacello databases. However, to avoid any confusion “Table 1” was omitted.
Point 7: To follow the discussion (e.g. ln 331 – 333), it would be helpful to have the information about the class of each chitinase gene in the Figures (Fig. 6 in this case).
Response 7: Information about the class of each chitinase gene was added to Figure 6.
Point 8: Ln. 346 ‘Class I chitinases have been extensively studies for’ - check English
Response 8: Grammatical error corrected.
Reviewer 2 Report
The manuscript by Bartholomew et al. reports an impressive and rigorous work. Chitinases appear as a good target for genetically improving the plant resistance against chitin-containing pathogens. The reported work builds the grounds for further important studies to control the resistance of cucumber to, for example, fusarium wilt.
The manuscript is well structured and well written.
Induction of defense responses in plants starts with a pathogen recognition step, followed by a complex cascade of signals, where the induction/repression of gene expression is the final step. In their discussion about differences in response between resistant and susceptible lines, it would be good for the authors to mention those upstream important steps. Chitinases are only one of the PR-proteins, so one element of the wall. How do the authors explain that many genes are not induced upon inoculation in the resistant line?
I recommend the manuscript to be accepted for publication after minor corrections, as listed below.
The text is too small in Figs 1, 2 and 6.
l. 21 and 22: it is not clear at that stage that the figures between brackets refer to the line number.
l. 31: not only fungi, all chitin-containing pathogens. It would be good as well to mention the structure of chitin
l. 87: the distinction of endo and exo-chitinase has not been introduced
l. 183: Which line has been used? Any detail about the tissue collected?
l. 238: Fig 6: it would be useful for the reader to have the class number added to the gene name. It would be useful to add in the fig captions which is line is the resistant and which one is the susceptible ones.
l. 251: which line was affected by those modifications?
l. 362-363: Do they appear in the identified response elements?
l. 475: …. chitinase “in the 3461 resistant line”, CsChi23, …
Author Response
Response to Reviewer 2 Comments
Point 1: Induction of defense responses in plants starts with a pathogen recognition step, followed by a complex cascade of signals, where the induction/repression of gene expression is the final step. In their discussion about differences in response between resistant and susceptible lines, it would be good for the authors to mention those upstream important steps. Chitinases are only one of the PR-proteins, so one element of the wall. How do the authors explain that many genes are not induced upon inoculation in the resistant line?
Response 1: Information related plant defense responses (detection, signaling and induction of defenses) and our explanation regarding genes not induced in resistant line upon inoculation were added to the discussion.
As a larger gene family, it is important to elucidate the multiple functions of chitinase genes. Chitinases not only play a major role in defense against fungal pathogens, but also have an important function in the regulation of growth and development in plants. We speculate that chitinase genes not induced upon inoculation in resistant line may be related to other functions. We also acknowledge the complexity of plant defense responses and many underlying factors such as hormone signaling and polymorphisms in chitinase gene and regulatory regions may affect gene expression.
Point 2: The text is too small in Figs 1, 2 and 6.
Response 2: The font sizes of Figures 1, 2 and 6 were increased. The image quality was also increased within the text and high-quality TIFF files (resolution 300 dpi) were uploaded along with the manuscript.
Point 3: l. 21 and 22: it is not clear at that stage that the figures between brackets refer to the line number.
Response 3: The cucumber line numbers were removed from the abstract and the terms “fusarium wilt-susceptible and -resistant lines” were used.
Point 4: l. 31: not only fungi, all chitin-containing pathogens. It would be good as well to mention the structure of chitin.
Response 4: Lines 32-33. The authors accepted the reviewer’s suggestions and made the required changes.
Point 5: l. 87: the distinction of endo and exo-chitinase has not been introduced
Response 5: These terms were introduced.
Point 6: l. 183: Which line has been used? Any detail about the tissue collected?
Response 6: Information about the line used and tissue collected was added.
C. sativus L. var. sativus cv. 9930 (same as the reference genome) was used for RNA-Seq. Tissues included leaf, stem, female flower, male flower, ovary, root, and tendril. Additional information added to the method section.
Point 7: l. 238: Fig 6: it would be useful for the reader to have the class number added to the gene name. It would be useful to add in the fig captions which is line is the resistant and which one is the susceptible ones.
Response 7: Information about the class of each chitinase gene was added to Figure 6.
In the figure caption, the terms “resistant and susceptible” were added.
Point 8: l. 251: which line was affected by those modifications?
Response 8: Cucumber line information and SNP locations were added.
Point 9: l. 362-363: Do they appear in the identified response elements?
Response 9: Genetic variations in the putative CsChi23 promoter fragment of line 3461 and 3229 was validated by cloning and sequencing. The impacts of these SNPs and InDels on cis-regulatory elements were predicted.
Point 10: l. 475: …. chitinase “in the 3461 resistant line”, CsChi23, …
Response 10: The authors accepted the reviewer’s suggestions and made the required changes.